# Research on Defect Detection in Kubo Peach Based on Hyperspectral Imaging Technology Combined with CARS-MIV-GA-SVM Method

**DOI:** 10.3390/foods12193593

**Published:** 2023-09-27

**Authors:** Lixiu Zhang, Pengcheng Nie, Shujuan Zhang, Liying Zhang, Tianyuan Sun

**Affiliations:** 1College of Biosystems Engineering and Food Science, Zhejiang University, Hangzhou 310058, China; z18503483861@163.com; 2College of Agricultural Engineering, Shanxi Agricultural University, Jinzhong 030800, China; nicosty@163.com; 3State Key Joint Laboratory of Environment Simulation and Pollution Control, School of Environment, Beijing Normal University, Beijing 100875, China; zlyzhangliying@mail.bnu.edu.cn

**Keywords:** Kubo peach, defect, CARS-MIV, GA-SVM, nondestructive testing

## Abstract

Due to the dark red surface of ripe fresh peaches, their internal injury defects cannot be detected using the naked eye and conventional images. The rapid and accurate detection of fresh peach defects can improve the efficiency of fresh peach classification. The goal of this paper was to develop a nondestructive approach to simultaneously detecting internal injury defects and external injuries in fresh peaches. First, we collected spectral data from 347 Kubo peach samples using hyperspectral imaging technology (900–1700 nm) and carried out pretreatment. Four methods (the competitive adaptive reweighting algorithm (CARS), the combination of CARS and the average influence value algorithm (CARS-MIV), the combination of CARS and the successive projections algorithm (CARS-SPA), and the combination of CARS and uninformative variable elimination (CARS-UVE)) were used to extract the characteristic wavelength. Based on the characteristic wavelength extracted using the above methods, a genetic algorithm optimization support vector machine (GA-SVM) model and a least-squares support vector machine (LS-SVM) model were used to establish classification models. The results show that the combination of CARS and other feature wavelength extraction methods can effectively improve the prediction accuracy of the model when the number of wavelengths is small. Among them, the discriminant accuracy of the CARS-MIV-GA-SVM model reaches 93.15%. In summary, hyperspectral imaging technology can accomplish the accurate detection of Kubo peaches defects, and provides feasible ideas for the automatic classification of Kubo peaches.

## 1. Introduction

“Early maturity Okubo Peach”, referred to as the “Kubo peach”, is a kind of premature variety of peach that originated in Japan, in Okubo; this fruit type has a fruit weight of about 230–280 g, dense meat, less fiber, and more juice; contains 16.48% soluble solid (16.48%); and is high-quality, rich in nutrition, and popular with consumers [1]. In the growing and harvesting process of Kubo peaches, due to climate and unavoidable collision in the harvesting process, the Kubo peach is prone to defects such as dark wounds, lesions, rot, scars, and marks [2,3]. The existence of these defects on the fruit surface will reduce the quality and market value of the fruit. When purchasing fruits, consumers tend to choose fruits with a good appearance. However, for obvious external obvious defects such as scab, rot, and others, peaches can be detected using conventional visual technology. While mature peaches are mostly red, the naked eye cannot identify damaged peaches, and ordinary machine vision cannot identify them, which leads to the classification and sorting efficiency of peaches not being high, affecting the commodity value of the fruit, and then, affecting the export and sales of Kubo peaches. At present, the peach fruit on the market mainly relies on manual grading. Manual classification has the problems of low efficiency and low accuracy [4]. Therefore, it is of practical value to study a fast and efficient method for the detection of dark wounds and obvious external defects in Kubo peaches.

Hyperspectral imaging is a non-contact image acquisition technology used to obtain continuous spectral information for objects in different wavelength ranges. It combines optical and digital image processing technology to provide rich spectral data and spatial information, making the detection and analysis of the surface composition, chemical composition, and specific characteristics of the object more accurate and comprehensive. Hyperspectral imaging combined with chemometric approaches is proven to be a powerful tool for the quality evaluation and control of fruits [5,6], as it enables the assessment of internal properties that cannot be inspected using computer vision, including soluble solid content [7], acidity [8], and texture [9]. Therefore, to detect fruit quality defects, such as immaturity [10], bruising [11], etc., it is feasible to use hyperspectral imaging technology to detect the dark wound defects and external defects of peaches.

In recent years, hyperspectral imaging technology has been widely applied to the external quality detection of fruits and vegetables, and the research subjects mainly include dates, cucumbers, cherries, citrus fruits, apples, peaches [12,13,14,15,16,17,18,19,20,21,22], etc. Most scholars have combined hyperspectral imaging technology with the related stoichiometry, and have obtained relatively scientific research results. In terms of internal quality detection, Li et al. [13] combined hyperspectral imaging technology with a multiple linear regression model to predict the soluble solid content of Hami jujubes, and the correlation coefficient of the prediction set reached 0.857. Li. et al. [14] combined hyperspectral technology with the CARS method to detect cucumber hardness and water loss, two indexes representing cucumber freshness. The final correlation coefficient of the PLSR model for hardness was 0.942, and that of the PLSR model for moisture was 0.822. Pullanagari et al. [15] combined hyperspectral imaging technology with a partial least-squares regression (PLSR) model and a Gaussian process regression (GPR) model to detect the hardness and total soluble solids of sweet cherries. Finally, the correlation coefficient of total soluble solids predicted using the GPR model was 0.88. The correlation coefficient of hardness prediction was 0.60. He et al. [16] combined hyperspectral imaging technology with multiplicative scattering correction, Savitzky–Golay, and first-order derivative pretreatment methods to establish a PLSR model to predict the water content of dried purple potatoes, and the correlation coefficient of the final model reached 0.975. In terms of defect detection, Tang et al. [17] fitted the hyperspectral imaging data of apples with different damage levels using piecewise nonlinear curves, studied the spectral data of damaged apples within the band range of 386–1016 nm, and concluded that the final score detection accuracy of this method reached 97.33%. Xu et al. [18] combined hyperspectral imaging technology with a partial least-squares regression model to detect the relationship between the damage degree and internal attribute quality of mangos, and graded mangos according to the damage degree. The final classification accuracy was 77.8%. WANG et al. [19] selected different pretreatment methods to establish an LS-SVM model based on the spectral data of the upper and lower surfaces of citrus leaves with yellow dragon disease. The results showed that the recognition rates of the upper and lower surfaces of citrus leaves were 100% and 92.5%, respectively, when the second derivative was selected as the pretreatment method. Zhang et al. [20] used mean-PC image, an improved watershed segmentation algorithm, and hyperspectral technology to distinguish normal oranges from defective oranges, and the overall classification accuracy reached 97.73%. Based on mean-PC5 and the simple global threshold method, rotten oranges and intact oranges were identified, and the recognition rate reached 100%. Zhang H. et al. [21] used hyperspectral imaging technology to detect scab, black spot, root rot, and brown disease in citrus, with a 94% final discrimination rate. Chen Si et al. [22] used hyperspectral imaging technology and the band selection method to propose defect region segmentation and recognition algorithms for peach brown rot and scab, with final recognition efficiencies of 96.9% and 88.4%, respectively.

These studies used hyperspectral imaging technology to model and analyze the internal qualities and external defects of fruits; they achieved good results with research methods primarily based on one or more single-feature wavelength extraction methods, which have wavelength redundancy problems or wavelengths that are too small. The wavelength combination method can maximize useful information from samples and improve detection accuracy. There are few reports regarding using the CARS-MIV combination method to extract characteristic wavelengths and establish GA-SVM and LS-SVM classification models to detect peach dark wound defects.

Based on the above problems, this study aimed to use hyperspectral imaging technology combined with stoichiometric methods to achieve rapid, efficient, nondestructive batch identification of the Kubo peach. This study’s specific process included the following: 

(1) Collecting Kubo peach fruit with different defect types and obtaining hyperspectral image data between 900 nm and 1700 nm; 

(2) Selecting the appropriate pretreatment method by establishing a partial least squares (PLS) model; 

(3) Introducing the CARS-MIV group method to extract the feature wavelength and compare the results with other classical feature wavelength extraction methods; 

(4) Introducing the GA-SVM discriminant model and comparing it with the classical LS-SVM model to select the best prediction model. 

In this study, HSI imaging technology provided a new research method for the rapid, efficient batch identification of Kubo peach external defects and the accurate classification of Kubo peach fruit.

## 2. Materials and Methods

### 2.1. Sample Collection

The sample consisted of Kubo peaches harvested on 12 July 2022, from Xi shan di Village, Jin zhong City, Shanxi Province, China. To ensure accuracy, the principles of uniform size (single fruit weight 170–180 g) and complete defect types (intact peaches, scab peaches, and rotten peaches) were selected during harvest. Damaged Kubo peach samples were placed in a paper box to simulate collision during harvest and transportation. After selection and analysis, 347 Kubo peaches were chosen for spectral data analysis. Selected Kubo peaches were left at room temperature for 24 h, and the three sample types were randomly divided into 259 correction sets and 88 prediction sets according to the Kennard–Stone algorithm ratio of 3:1. Figure 1 shows sample images of four Kubo peach types.

### 2.2. Hyperspectral Imaging System

Hyperspectral imaging technology combines spectral information with spatial information [23]. It has hyperspectral and spatial resolution, and higher resolution and accuracy for the surface components, component distribution, and feature extraction of objects [24]. The hyperspectral sorting instrument adopted in this study is shown in Figure 2, and the specific parameters of the hyperspectral sorting instrument are shown in Table 1.

As the spectrum was affected by changes in light intensity and the dark current in the lens [25] during the acquisition process, the acquired images were not clear, so it was necessary to perform black-and-white correction before spectrum acquisition [26].

The following calculation formula was used for black-and-white correction.
(1)$$R=\frac{IR-ID}{IW-ID}$$

Note: *R*—corrected image; *IR*—original image; *ID*—blackboard correction image; *IW*—whiteboard correction image.

### 2.3. Main Data Processing Methods

#### 2.3.1. CARS-MIV Feature Variable Extraction Method

The competitive adaptive reweighting algorithm (CARS) is an unstable algorithm; the regression coefficient R of the variable set changes as the sample numbers change [27]. Therefore, the absolute value of the regression coefficient R cannot effectively reflect a variable’s importance, and the feature variable extracted by a single CARS algorithm will occupy approximately 1/3 of the spectral data’s full wavelength. Modeling data are not concise enough. Therefore, the average influence value algorithm (MIV) [28] was introduced in this study to conduct secondary screening of characteristic wavelengths screened using the CARS algorithm, further simplifying the data and improving the model’s accuracy. The average impact value algorithm (MIV) was used for feature selection. It evaluated the features’ importance by calculating their average influence value on the predicted results, and then, carried out feature selection. The MIV algorithm process was as follows (Figure 3):

Step 1: Use the original sample set to establish the Shenjing network training model Svmtrain.

Step 2: Increase and decrease the original P1 and P2 by 10%, respectively, to form new training set samples, P1 and P2.

Step 3: Use the new training set samples, P1 and P2, as the neural network’s input model, and obtain two output results, y1 and y2.

Step 4: The difference between the two output results (y1–y2) is the impact value (IV) on the output variable after changing the original input variable.

Step 5: Average the IV value according to the number of observed cases to obtain the MIV of the input to the output variable. A loop structure is used to successively calculate each input variable’s MIV, and the characteristic wavelength variables whose contribution rates are greater than 95% are screened out according to the descending order of the absolute value.

#### 2.3.2. GA-SVM Modeling Method

A support vector machine (SVM) is a powerful classification and regression model; however, it also has some disadvantages. An SVM has relatively low processing efficiency of large-scale data sets and high-dimensional data, presents difficulty when selecting the appropriate kernel function for nonlinear problems, requires selection of the appropriate penalty parameter C, etc. As a classical optimization algorithm, a genetic algorithm has good applicability, a fast search speed, and high efficiency [29]. In this study, a genetic algorithm (GA) was used to optimize the C and r parameters of the support vector machine (SVM). The steps based on the GA-SVM [30] algorithm are as follows; the process is shown in Figure 4.

Step 1: Set the parameters and reasonably select the genetic algorithm’s operating parameters and the SVM’s parameter optimization range.

Step 2: Form the initial population, convert the kernel function parameter r and SVM penalty factor C into genotype data with genetic characteristics via binary coding; randomly generate the initial population.

Step 3: Calculate each individual’s fitness to evaluate their importance.

Step 4: Use selection, crossover, and mutation operations in the genetic algorithm to optimize parameters c and g to determine whether each individual’s fitness meets the termination criteria. If yes, perform Step 6; otherwise, perform Step 5.

Step 5: Select individuals with high fitness for crossover and mutation operations to form a new population. Return to Step 3 until the stop criteria are met, ending the loop.

Step 6: Parse the code and output the parameter value.

Step 7: Apply optimal parameters to the SVM model for solution analysis.

### 2.4. Model Evaluation Index

In this study, five indexes (namely, the coefficient of determination of the training set (RC2), the root-mean-square error (RMSEC), the coefficient of determination of the test set (RP2), the root-mean-square error of the training set (RMSEP) and the discrimination accuracy of the prediction set) were used to comprehensively evaluate the model. The closer the correlation coefficient between the correction set and the prediction set is to 1, the closer the root-mean-square error is to 0, and the higher the discrimination accuracy is, the better the model is considered to be.

## 3. Results and Discussion

### 3.1. Average Spectral Curves of Four Types of Kubo Peach Samples

The ROI region selection should best represent image content features; this can help reduce the calculation complexity, improve the algorithm’s performance, and provide increased accuracy [31]. In this study, the ROI region in the center of the equator of intact peach samples and ROI regions in the defects of damaged peaches, scab peaches, and scarred peaches were selected. After sorting the spectral data from the four sample types, average spectral curves for each of the four sample types were drawn using Origin8.5 software, as shown in Figure 5.

As can be seen in Figure 5, there were significant differences in the average spectral curves of the four Kubo peach sample types. This was due to differences in the skin thickness and spectral reflectance of intact and defective peaches. Wave peaks at 1100 nm and 1290 nm of the four sample types were related to the Kubo peach’s excessive reflectivity, and troughs at 1190 nm and 1450 nm were related to the Kubo peach’s internal water and sugar absorption, which are O-H third-order and second-order frequency-doubling characteristic absorption peaks [32]. The four sample types’ overall reflectance showed a decreasing trend.

### 3.2. Spectrum Pretreatment

Spectral pre-processing refers to a series of processing operations on acquired spectral data to extract useful information or improve data quality [33]. In this study, derivative-gap-segment (D-GS), standard normalization (SNV), baseline, and a smoothing-median filter (SMF) were used. Derivative Savitzky–Golay (D-SG) and five pretreatment methods were used to process the original spectral data, and PLS models were established. The prediction results are shown in Table 2.

As shown in Table 1, the model pre-treated using DSG had relatively high accuracy and a relatively low standard deviation. Its correction set determination coefficient was 0.75 with a standard deviation was 0.56, and its prediction set determination coefficient was 0.85 with a standard deviation was 0.45. Therefore, data pre-processed using DSG were selected for the follow-up study.

### 3.3. Selection of Characteristic Wavelength Variables

#### 3.3.1. CARS Competitive Adaptive Weighting Algorithm

The CARS competitive adaptive reweighting algorithm is a sampling algorithm based on adaptive weights, and is often used to solve data imbalance problems. In data imbalance problems, the training set has a large difference in the number of samples from different classes, which may cause the classifier to tend to predict classes that occur more frequently, while the prediction effect is poor for rare classes. The CARS algorithm aims to solve this problem. By adaptively adjusting sample weights and iteratively updating sample weights and model parameters, the CARS algorithm can achieve a state of dynamic balance and improve the prediction accuracy of rare categories [34]. The CARS feature wavelength extraction process is illustrated in Figure 6.

Pre-processed spectral characteristic variables were extracted based on the CARS algorithm, and the number of Monte Carlo samplings was set to 50. Figure 6a shows the attenuation function selection process; the number of variables showed a trend of rapid decline at first, and then, a steady decline with an increase in sampling times. Figure 6b shows that the root-mean-square error of cross-validation first slowly decreased, and then, rapidly increased with an increase in sampling times. Figure 6b shows that the root-mean-square error of cross-validation (RMSE) was smallest at the 25th sampling time, and its minimum value was 0.6190. Figure 6c shows the change process of the regression coefficient’s path value with an increase in sampling times. A total of 24 characteristic wavelengths were screened out by running the CARS algorithm 25 times.

#### 3.3.2. Feature Wavelength Extraction Using the CARS-MIV Method

The average influence value (MIV) algorithm can screen out variables with high correlation for modeling analysis. In this study, 24 feature variables were screened using the CARS algorithm, and feature variables with contribution rates greater than 95% were screened using the MIV algorithm to achieve a secondary screening, so that the feature wavelength variables were more representative. Selected wavelengths were sorted in descending order, and 11 characteristic wavelength variables were finally selected for subsequent modeling analysis. The MIV screening process is shown in Figure 7, the selected characteristic wavelength is shown in Figure 8, and the cumulative contribution rate of the characteristic wavelength is shown in Figure 9.

#### 3.3.3. Feature Wavelength Extraction Using the CARS-SPA Method

There was a data imbalance problem among the 24 characteristic-wavelength-adjacent variables formed using the CARS algorithm. Therefore, the SPA method of screening characteristic wavelengths was used to reduce the dimension twice. Data imbalances in CARS extraction variables could be eliminated. The variation range of the number of characteristic spectrum variables was set to 1–25. The SPA algorithm reduced the dimensions of 24 characteristic spectra, and seven characteristic variables were extracted when the RMSEC was 0.62033. The characteristic wavelength variables are shown in Figure 10.

#### 3.3.4. Feature Wavelength Extraction Using the CARS-UVE Method

The uninformative variable elimination (UVE) algorithm is a band selection algorithm based on the regression coefficient of the PLS model. The value of the regression coefficient is used to measure whether or not a variable is valid. A new matrix can be obtained by randomly setting a noise matrix and placing it after the spectral matrix. The PLS regression process is performed on the new matrix using the leave-one-interaction-out verification method. The reliability of each variable can be obtained by comparing the PLS results with the standard deviation and average value of the regression coefficients. Then, according to the reliability value, feature wavelengths with large reliability values are screened out [35]. The UVE algorithm can delete uninformative variables in the spectral information, but when the number of bands is large, the number of feature wavelengths selected by the algorithm remains large, which is not conducive to the development of spectral equipment. Therefore, the UVE algorithm is often combined with other algorithms to further screen feature variables [36]. In this study, the CARS algorithm was combined with the UVE algorithm. On the basis of 24 feature variables screened using the CARS algorithm, the UVE algorithm was used to screen 15 feature wavelengths twice. The filtering process is shown in Figure 11.

The dotted blue horizontal lines in Figure 11 are the upper and lower limits of the thresholds, and the variables between these lines are information unrelated to the Kubo peach classification prediction, which require deletion. The 15 variables outside the dotted line are the selected feature variables.

### 3.4. Modeling Results and Analysis

#### 3.4.1. Least-Squares Support Vector Machine (LS-SVM) Model

The principle of the LS-SVM modeling method is to take RBF as the model’s kernel calculation function, use a grid-search method on the basis of cross-validation, and adopt a global optimization method for the main SVM target parameters γ and δ^2^. The principle [37] that minimizes the root-mean-square error RMSEC was used to optimize the target parameters’ design.

#### 3.4.2. Support Vector Machine (GA-SVM) Parameter Optimization Model Based on the Genetic Algorithm

The genetic algorithm is an optimization algorithm based on the survival of the fittest and the natural selection mechanism in biological evolution. It has the characteristics of strong fitness, high optimization efficiency, and fast searching ability. In the GA-SVM model used in this study, the maximum number of genetic iterations was set to 100, the population size was set to 20, and the method of five-fold cross-validation was adopted to obtain the optimal penalty factor (cost) as C and core parameter (gamma) as γ. In this study, the feature wavelength sample data extracted using the four methods were normalized. After the initial training was complete, training sample data were input into the GA-SVM model for model training, and then, the prediction set sample was input into the trained model to obtain the training results. By comparing the predicted training results with the real predicted results, confusion matrices of the discriminant samples were obtained.

The prediction data results of the two models established using the four feature wavelength extraction methods are shown in Table 3, Table 4, Table 5 and Table 6.

According to the discrimination results presented in Table 3, Table 4, Table 5 and Table 6, the discrimination rate accuracy of the GA-SVM model established using the four methods of extracting characteristic wavelengths was much higher than that of the LS-SVM model. Therefore, the GA-SVM model was chosen as the optimal discrimination model. The comprehensive discrimination rates of GA-SVM models established using the CARS, CARS-SPA, and CARS-UVE methods were 89.11%, 88.73%, and 89.75%, respectively. The comprehensive discriminant rate of the GA-SVM model established using the CARS-MIV method was 93.15%, and it had the best discriminant effect. Therefore, the CARS-MIV-GA-SVM model was chosen as the optimal discrimination model. The highest discriminant rate of an intact peach was 100%. One scab peach was misjudged as a dark wound because some scab peaches have a certain degree of dark wound inside, which is not visible to the human eye, so miscalculations can occur. One black peach was misjudged as a scab peach; due to the depth of the black wound, the sample’s color characteristics showed obvious changes, resulting in machine identification as a scab peach. One rotten peach was misjudged as a scab peach; its rotten area was too small, resulting in machine identification as a scab peach. The GA-SVM model’s optimization process and prediction results established using four feature wavelength extraction methods are shown in Figure 12, Figure 13, Figure 14 and Figure 15, respectively.

### 3.5. Discussion

In this experiment, we applied the CARS-MIV combination method to extract feature wavelengths and establish a detection model capable of simultaneously evaluating the internal and external qualities of Kubo peaches. Spectral data pre-processed using DSG were screened using four characteristic wavelength extraction methods: CARS, CARS-MIV, CARS-SPA, and CARS-UVE. Compared with other feature variable screening methods, the two models based on the CARS-MIV method had higher discriminant rates. Based on the principle of the feature variables’ importance, the CARS-MIV method was used as a secondary screening method after feature variables were screened based on CARS. First, a neural network training model was established according to selected characteristic variables, and a new independent variable was formed by adding and reducing 10%, respectively, based on the original independent variable; the difference value could be calculated by inputting the original independent variable into the trained neural network model to obtain the variable‘s influence change value. According to this principle, the importance of the selected variable was screened twice. Important wavelength variables can be screened more accurately. Xu L. [38] used hyperspectral imaging technology to nondestructively detect kiwi fruit’s sugar content, a single feature wavelength extraction method, and a combination of multiple feature wavelengths to extract the feature wavelength, and then, carried out a comparative analysis. The research results showed that the model established using (CARS + IRIV)-SPA to extract the feature wavelength had the best prediction effect. This shows that it is feasible to use hyperspectral technology and a composition method to extract characteristic wavelengths and establish models to predict sugar content. Compared with the above studies, this experiment’s combination algorithm added a step to sort according to wavelength importance, so that feature variables with higher importance could be screened more accurately.

GA-SVM is also an optimization model that uses the classical genetic algorithm (GA) to optimize the C and γ parameters of SVM. Su J. [39] used the GA-SVM model formed using the GA optimization of SVM parameters to classify dried jujube varieties. Their research results showed that, compared with the traditional SVM model, the classification accuracy of the GA-SVM model was improved by more than 20%. Dan S. [40] used near-infrared spectroscopy and the GA-SVM model to study citrus-producing areas in 16 regions; their research results showed that the GA-SVM model’s recognition rate for citrus-producing areas reached 94.58%. Compared with the above studies, in this study, the GA-SVM model was used for the first time to distinguish the external quality classification of the Kubo peach. The results presented in Table 2, Table 3, Table 4 and Table 5 show that the discrimination rate accuracy of the GA-SVM model established using the four methods of extracting characteristic wavelengths was much higher than that of the LS-SVM model. Therefore, the GA-SVM model was chosen as the optimal discrimination model. The comprehensive discrimination rates of the GA-SVM model established using CARS, CARS-SPA, and CARS-UVE were 89.11%, 88.73%, 89.75%, and 88.59%, respectively. The comprehensive discriminant rate of the GA-SVM model established using CARS-MIV was 93.15%, and its discriminant effect was the best. Therefore, the CARS-MIV and GA-SVM models were combined to determine the Kubo peach’s external quality.

In this study, we applied this model to identify the external quality of only one peach variety; it is not universal in the detection of the external quality of other peach varieties. Therefore, in a future study, we will use more fruit varieties to realize the model’s potential detection capabilities, and to improve its universality.

## 4. Conclusions

To quickly detect Kubo peach internal injury defects and realize their accurate classification, Kubo peach internal and external defects were studied using hyperspectral imaging technology. A variety of pretreatment methods were used to process original spectral data, and the optimal pretreatment method was selected. The MIV algorithm was used to double-screen feature variables extracted using CARS according to their importance, GA was used to optimize the C and γ parameters of SVM to obtain the GA-SVM optimization model, and the optimized algorithm was compared with a single algorithm. Finally, it was concluded that combining CARS-MIV feature wavelength extraction with the GA-SVM model had the highest discrimination accuracy, optimized the external quality detection model for the Kubo peach, and provided a new research idea for the realization of Kubo peach quality detection. The conclusions drawn are as follows:

(1) The DSG spectral pretreatment method could better optimize spectral data, and spectral data after pretreatment could establish the PLS model with relatively high accuracy and low standard deviation. The correction set determination coefficient Rc2 was 0.7521, and the root-mean-square error RMSEC was 0.5693. The prediction set determination coefficient Rp2 was 0.8463, and the root-mean-square error RMSEP was 0.4563. Therefore, the data pre-processed using DSG were selected for the follow-up study.

(2) The data pre-treated using DSG were modeled and analyzed using four feature wavelength screening methods, namely, CARS, CARS-MIV, CARS-SPA, and CARS-UVE. The established GA-SVM model had a better discriminative effect than the LS-SVM model. Compared with the feature wavelength screening and model building methods, the GA-SVM model based on CARS-MIV-extracted feature wavelength data had the best prediction effect, and its discrimination accuracy reached 93.15%. Therefore, the CARS-MIV-GA-SVM model was selected as the optimal model to detect the Kubo peach’s external quality.

In summary, we proposed a detection method based on hyperspectral imaging technology to simultaneously detect the internal injury and external defects of fresh Kubo peaches. We introduced the idea of extracting feature wavelengths by combining CARS-MIV, CARS-SPA, CARS-UVE, and other methods. We combined the feature wavelength algorithm and the optimization model (GA-SVM) to enhance the classification model’s accuracy. This approach addressed the limitations of existing peach grading methods that solely rely on appearance and overlook internal damage in peaches.

## Figures and Tables

**Figure 1 foods-12-03593-f001:**
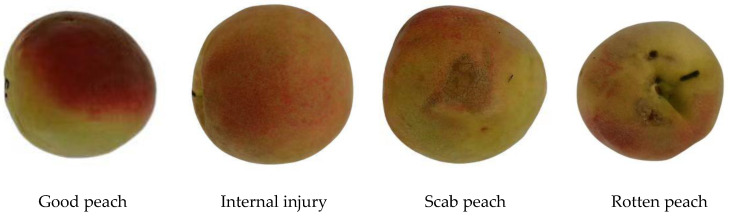
Sample images of an intact peach and defective peaches.

**Figure 2 foods-12-03593-f002:**
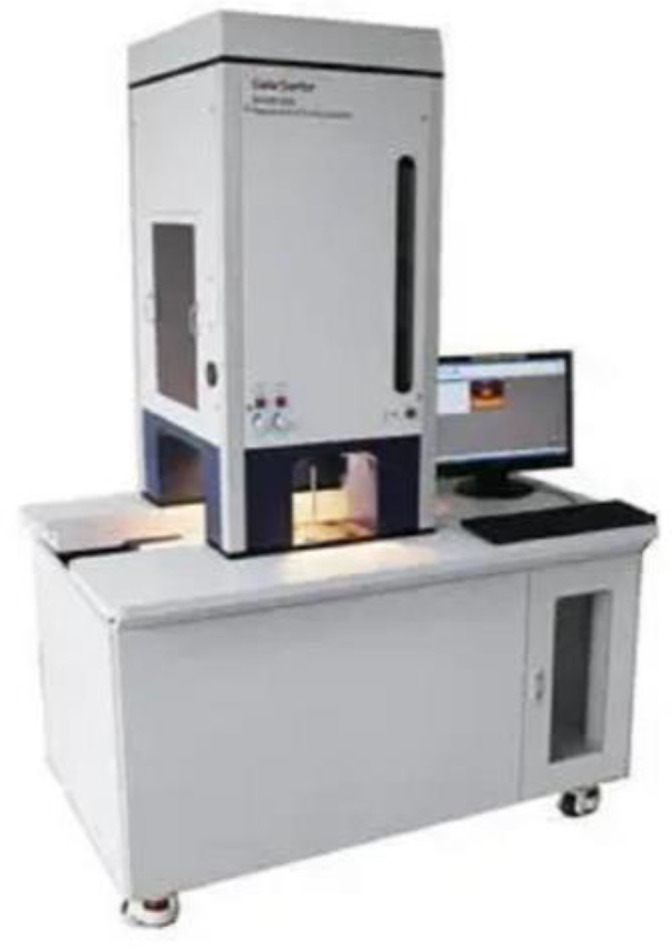
Hyperspectral sorter.

**Figure 3 foods-12-03593-f003:**
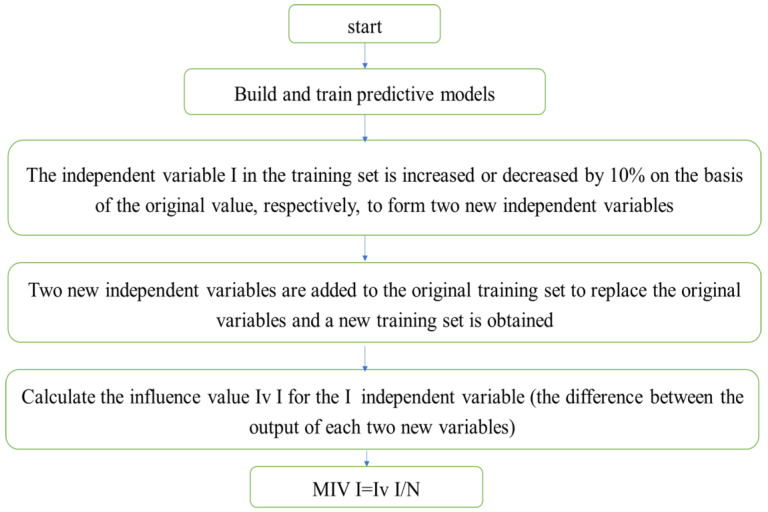
MIV algorithm flowchart.

**Figure 4 foods-12-03593-f004:**
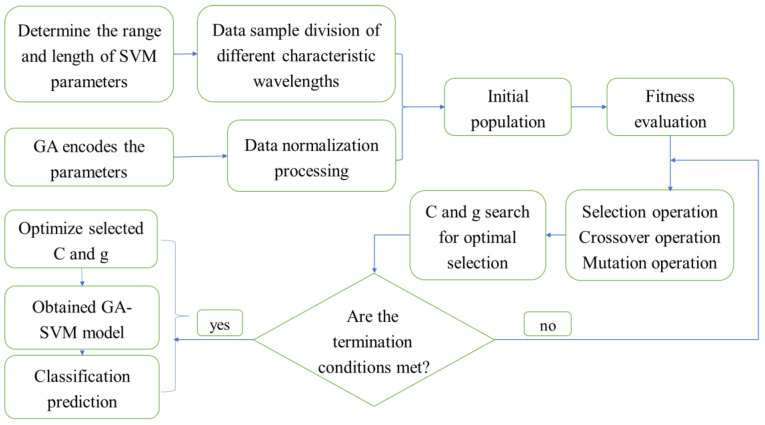
GA-SVM algorithm flowchart.

**Figure 5 foods-12-03593-f005:**
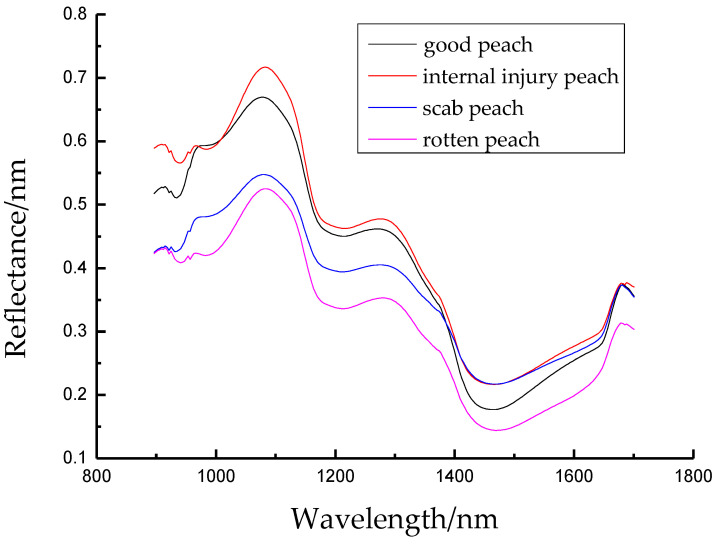
Average spectral curves of four sample types.

**Figure 6 foods-12-03593-f006:**
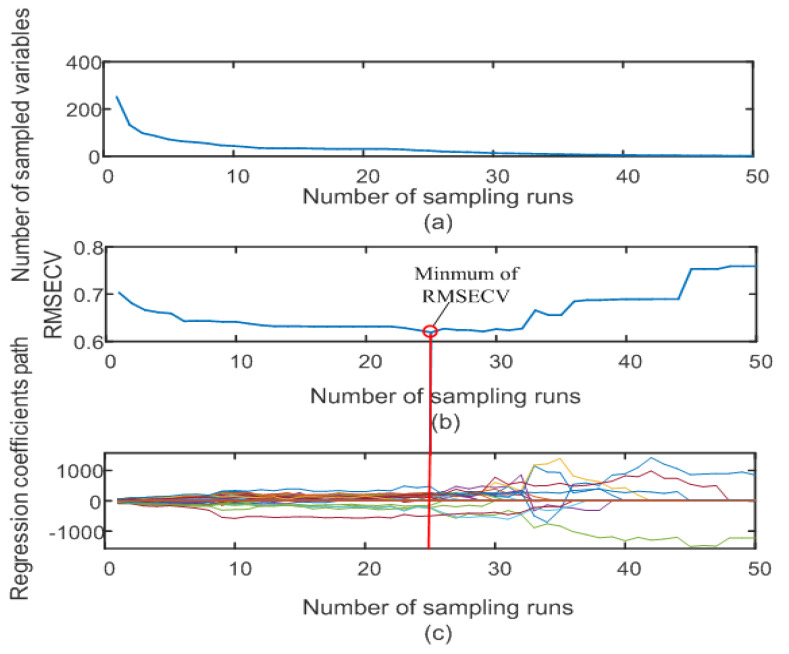
CARS feature wavelength extraction process. Variation of (**a**) the number of variables in the attenuation function, (**b**) the root-mean-square error value of cross-validation (**c**) the path value of the regression coefficient with the increase of the sampling number.

**Figure 7 foods-12-03593-f007:**
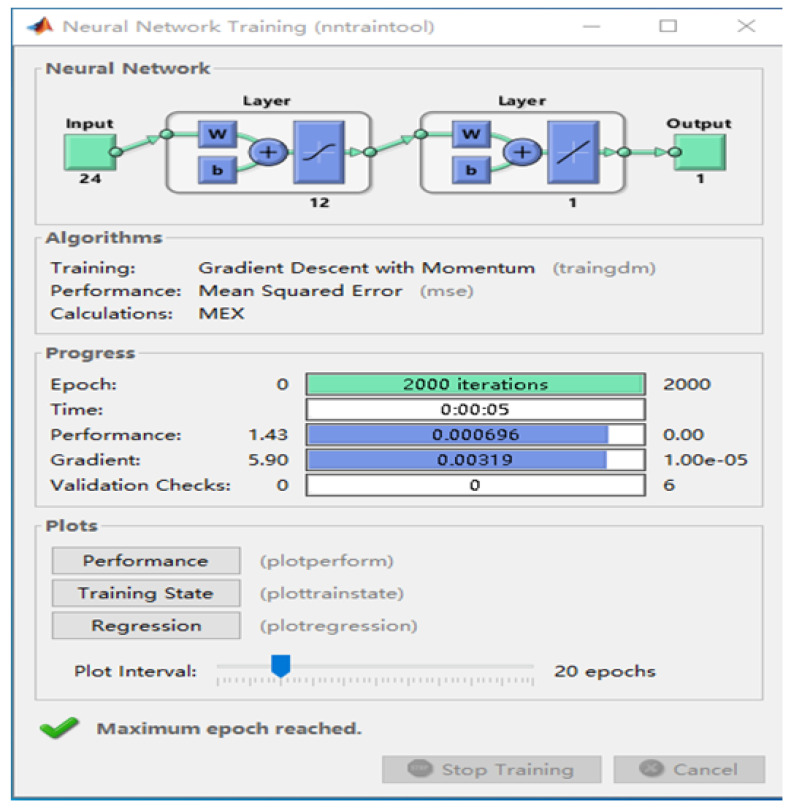
Wavelength extraction process of MIV secondary screening.

**Figure 8 foods-12-03593-f008:**
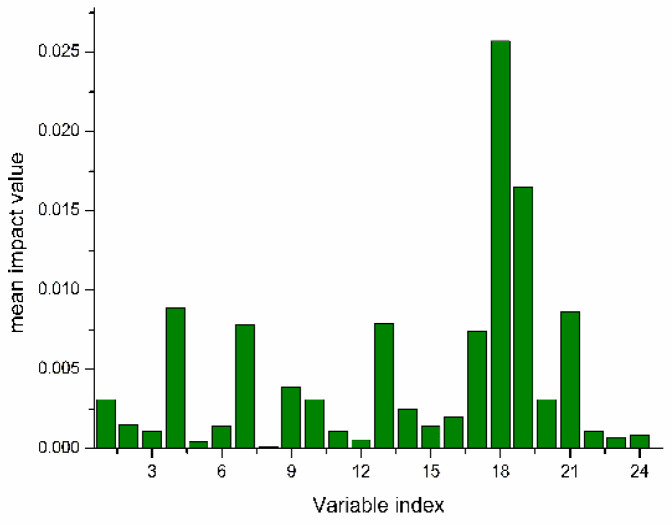
CARS-MIV feature variables’ values.

**Figure 9 foods-12-03593-f009:**
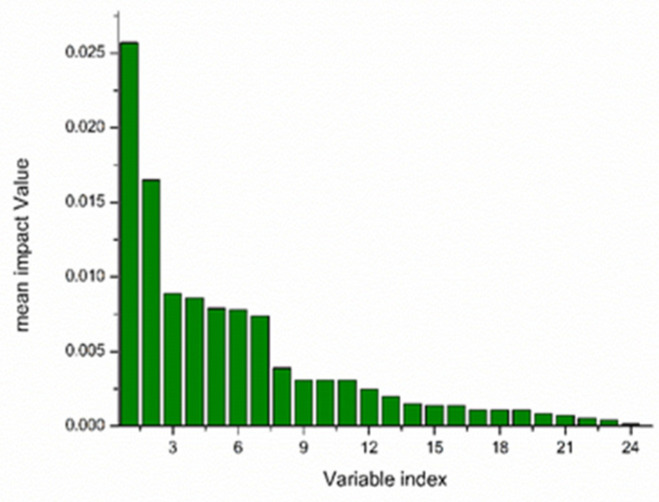
Importance ranking of CARS-MIV feature variables.

**Figure 10 foods-12-03593-f010:**
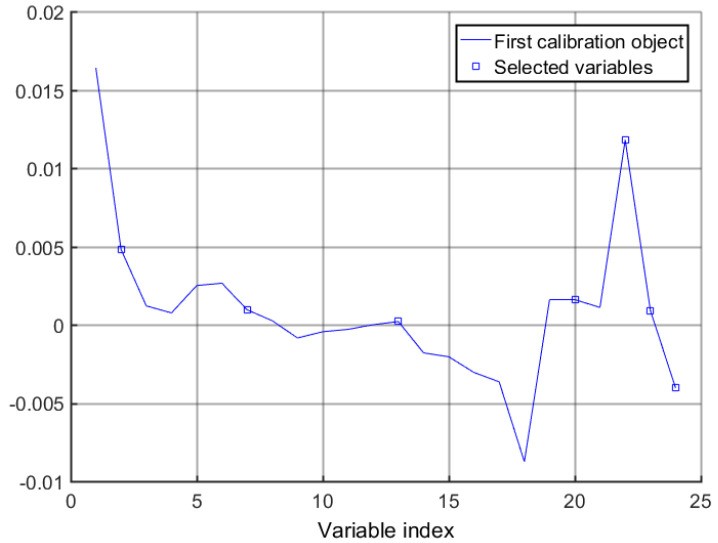
Distribution of CARS-SPA characteristic variables.

**Figure 11 foods-12-03593-f011:**
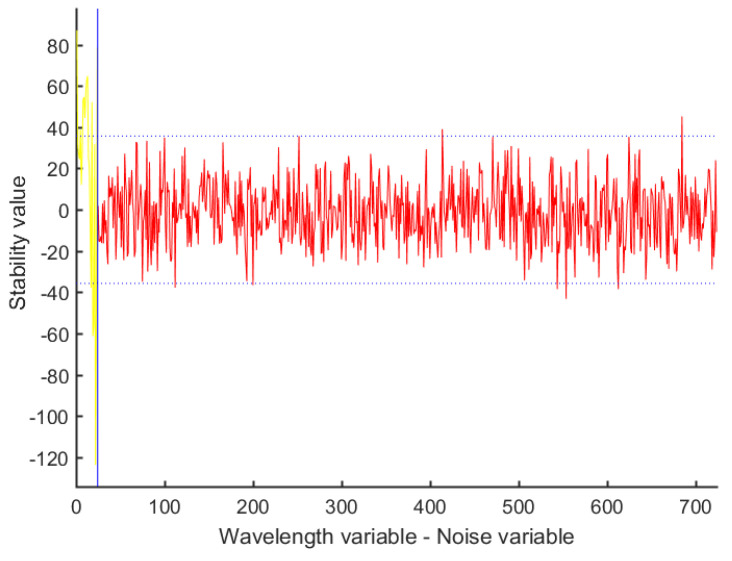
Feature wavelength extracted using the CARS-UVE method.

**Figure 12 foods-12-03593-f012:**
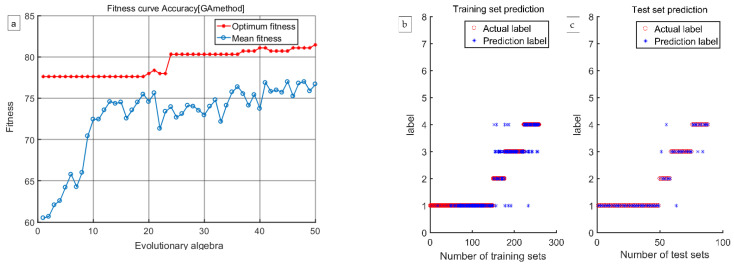
The CARS-GA-SVM model’s optimization process and prediction results: (**a**) the optimization process, (**b**) training results, and (**c**) test results.

**Figure 13 foods-12-03593-f013:**
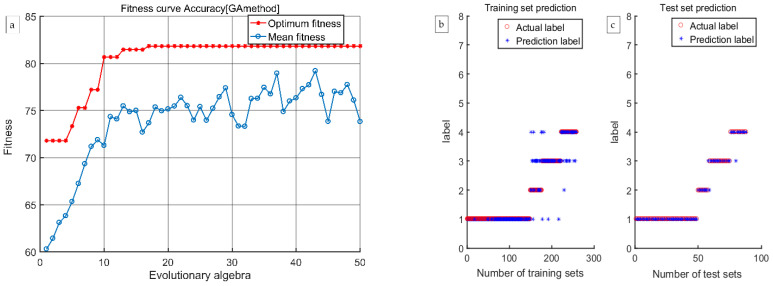
The CARS-MIV-GA-SVM model’s optimization process and prediction results: (**a**) the optimization process, (**b**) training results, and (**c**) test results.

**Figure 14 foods-12-03593-f014:**
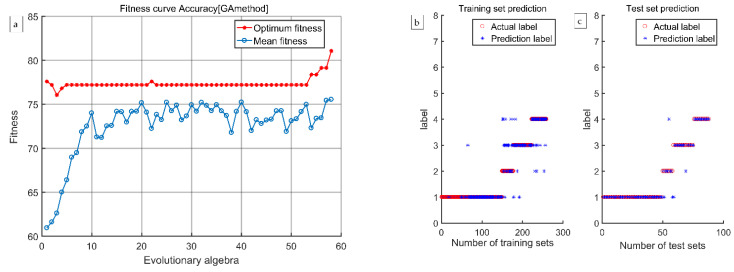
The CARS-SPA-GA-SVM model’s optimization process and prediction results: (**a**) the optimization process, (**b**) training results, and (**c**) test results.

**Figure 15 foods-12-03593-f015:**
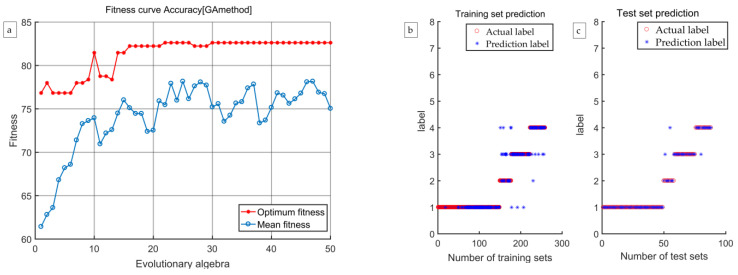
The CARS-UVE-GA-SVM model’s optimization process and prediction results: (**a**) the optimization process, (**b**) training results, and (**c**) test results.

**Table 1 foods-12-03593-t001:** Hyperspectral sorter’s parameters.

Name	Parameter
Model number	ZOLIX Gaia Sorter-type “Gaia” hyperspectral sorter
Main instrument parts	Image-lambda-n17e spectral camera, camera obscura, electronic control platform, bromo-tungsten lamp (4), external computer
Sample exposure time	20 ms
Lens distance from sample height	22 cm
Forward speed of the electronically controlled mobile platform	2 cm/s

**Table 2 foods-12-03593-t002:** PLS models established using different pretreatment methods.

Pretreatment Method	Correction Set		Prediction Set	
	RC2	RMSEC	RP2	RMSEP
Primary spectrum	0.72	0.60	0.86	0.45
DGS	0.74	0.57	0.83	0.47
DSG	0.75	0.56	0.85	0.45
Baseline	0.71	0.61	0.82	0.51
SMF	0.72	0.60	0.85	0.45
SNV	0.65	0.67	0.72	0.62

**Table 3 foods-12-03593-t003:** Confusion matrices of CARS-GA-SVM and CARS-LS-SVM models.

GA-SVM				Model	LS-SVM			
Good Peach	Internal Injury Peach	Scab Peach	Rotten Peach	Classification	Good Peach	Internal Injury Peach	Scab Peach	Rotten Peach
49	0	0	0	**Good** **peach**	49	0	0	0
0	7	1	1	**Internal injury peach**	0	4	5	0
0	0	16	1	**Scab** **peach**	0	6	11	0
0	0	2	11	**Rotten** **peach**	0	0	3	10
49	9	17	13	**Total**	49	9	17	13
100	77.7	94.12	84.62	**Discriminant** **Rate**	100	44.44	64.71	76.92
89.11				**Total** **Discriminant** **rate**	71.52			

**Table 4 foods-12-03593-t004:** Confusion matrices of CARS-MIV-GA-SVM and CARS-MIV-LS-SVM models.

GA-SVM				Model	LS-SVM			
Good Peach	Internal Injury Peach	Scab Peach	Rotten Peach	Classification	Good Peach	Internal Injury Peach	Scab Peach	Rotten Peach
49	0	0	0	**Good** **peach**	49	0	0	0
0	8	1	0	**Internal injury peach**	0	4	5	0
0	1	16	0	**Scab** **peach**	0	5	12	0
0	0	1	12	**Rotten** **peach**	0	0	3	10
49	9	17	13	**Total**	49	9	17	13
100	88.9	91.4	92.3	**Discriminant** **Rate**	100	44.44	70.59	76.92
93.15				**Total** **Discriminant** **rate**	71.52			

**Table 5 foods-12-03593-t005:** Confusion matrices of CARS-SPA-GA-SVM and CARS-MIV-LS-SVM models.

GA-SVM				Model	LS-SVM			
Good Peach	Internal Injury Peach	Scab Peach	Rotten Peach	Classification	Good Peach	Internal Injury Peach	Scab Peach	Rotten Peach
49	0	0	0	**Good** **peach**	49	0	0	0
2	6	0	1	**Internal injury peach**	0	4	5	0
1	1	15	0	**Scab** **peach**	0	6	11	0
0	0	0	13	**Rotten** **peach**	0	0	2	11
49	9	17	13	**Total**	49	9	17	13
100	66.67	88.24	100	**Discriminant** **Rate**	100	44.44	64.71	84.62
88.73				**Total** **Discriminant** **rate**	73.44			

**Table 6 foods-12-03593-t006:** Confusion matrices of CARS-UVE-GA-SVM and CARS-MIV-LS-SVM models.

GA-SVM				Model	LS-SVM			
Good Peach	Internal Injury Peach	Scab Peach	Rotten Peach	Classification	Good Peach	Internal Injury Peach	Scab Peach	Rotten Peach
49	0	0	1	**Good** **peach**	49	0	0	0
0	6	2	0	**Internal injury peach**	0	7	2	0
0	0	17	12	**Scab** **peach**	0	4	13	0
0	0	1	13	**Rotten** **peach**	0	0	4	9
49	9	17	13	**Total**	49	9	17	13
100	66.67	100	92.31	**Discriminant** **rate**	100	77.77	76.47	69.23
89.75				**Total** **Discriminant** **rate**	80.87			

Note: This is an example to explain the methods listed in the table: CARS-MIV means that the average influence value MIV algorithm was used to reduce the dimensions of 24 spectral variables extracted using the CARS method twice.

## Data Availability

The data used to support the findings of this study can be made available by the corresponding author upon request.

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
