# Peer review of "Research on Defect Detection in Kubo Peach Based on Hyperspectral Imaging Technology Combined with CARS-MIV-GA-SVM Method"

_foods, 2023, doi:10.3390/foods12193593_

Round 1

Reviewer 1 Report

Present paper entitled "Research on Defect detection of Kubo peach Based on Hyperspectral Imaging Technology combined CARS-MIV-GA-SVM method' is well structured and the topic is intersting but there are few comments:

In line 124, it is stated that previous research used only one feature for modeling, while there are several hybrid papers in hyperspectral imaging such as

first_page

settings

Order Article Reprints

·         Comparison of Classic Classifiers, Metaheuristic Algorithms and Convolutional Neural Networks in Hyperspectral Classification of Nitrogen Treatment in Tomato Leaves

The authors claimed that it is not possible to identify some external defects in ripe peaches using machine vision, but they did not illustrate any images to prove this claim. Because in Figure 1, the identification is clear and this claim is rejected

The claimed innovation (identifying the external defects of peaches is challenging) was not convincing to me. Please give a convincing reason

The graphs do not have a good resolution. Please correct it

State the research perspective in the form of future work

Author Response

Research on Defect detection of Kubo peach Based on Hyperspectral Imaging Technology combined CARS-MIV-GA-SVM method

Reviewer: 1 - Comments and Suggestions for Authors

Present paper entitled "Research on Defect detection of Kubo peach Based on Hyperspectral Imaging Technology combined CARS-MIV-GA-SVM method' is well structured and the topic is interesting but there are few comments. Some specific comments towards the current manuscript are listed below.

Reply:

Thank you for reviewer’s comments. We sincerely appreciate this reviewer’s efforts in reviewing our manuscript. We have revised the manuscript according to the reviewer’s suggestions which certainly have improved the quality of the article. The revised parts are highlighted in yellow in the revised manuscript. Detailed responses are given below.

1.In line 124, it is stated that previous research used only one feature for modeling, while there are several hybrid papers in hyperspectral imaging such as First page settings Comparison of Classic Classifiers, Metaheuristic Algorithms and Convolutional Neural Networks in Hyperspectral Classification of Nitrogen Treatment in Tomato Leaves.

Reply:

Thank you for your suggestion. It was the author's carelessness that did not to Express clearly. The revised sentence is:

The above research mainly used one or more single feature wavelength extraction methods, which has the problem of wavelength redundancy or too little. Wavelength. Combination method can maximize the useful information of samples and improve the detection accuracy. (Lines:219-222) Therefore, the characteristic wavelength combination method is adopted in this paper.

2.The authors claimed that it is not possible to identify some external defects in ripe peaches using machine vision, but they did not illustrate any images to prove this claim. Because in Figure 1, the identification is clear and this claim is rejected

Reply:

Thank you for your suggestion. It was the author's miswritten that did not to Express clearly.

My explanation is as follows:

Machine vision technology can detect obvious external defects of ripe peaches, such as scarred peaches, insect-bitten peaches, bird pecking peaches, etc., but it cannot detect the internal injury defects of ripe peaches (Internal injury defects: during the picking and transportation process, the peach skin remains unbroken, but the fruit sustains a certain degree of bruising inside.), which cannot be observed by human eyes and cannot be detected by machine vision. While the hyperspectral technology can detect the internal characteristics of fruits, such as immaturity [10], bruising [11], contamination, frostbite and other defects. (Lines:119-123)

Figure 1 shows each sample diagram, from left to right: good peach, internal injury peach, scab peach, rotten peach. As we can see from Figure 1, there is no difference in surface between the internal injury peach and the good peach. Therefore, we say that machine vision cannot identify the internal injury defect of peaches.

Good peach

Internal injury peach

Scab peach

Rotten peach

Figure 1. Sample diagram of intact peach and defective peach

3.The claimed innovation (identifying the external defects of peaches is challenging) was not convincing to me. Please give a convincing reason.

Reply:

Thank you for your suggestion. My explanation is as follows:

First there is literature: hyperspectral imaging combined with chemometric approaches is proven to be a powerful tool for the quality evaluation and control of fruits [5,6] as it enables the assessment of internal properties that cannot be inspected with computer vision, including soluble solid content [7], acidity [8], texture [9]. To detect fruit quality defects, such as immaturity [10], bruising [11] etc. (Lines:119-123)

So, the innovations in this paper are as follows:

We proposed a detection method based on hyperspectral imaging technology to simultaneously detect the internal injury and external defects of fresh peaches. We introduced the idea of extracting feature wavelengths by combining CARS-MIV, CARS-SPA, CARS-UVE, and other methods. We combined the feature wavelength algorithm and the optimization model (GA-SVM) to enhance the accuracy of the classification model. This approach addresses the limitations of existing peach grading methods that solely rely on appearance and overlook the internal damage of peaches. (Lines:550-557)

4.The graphs do not have a good resolution. Please correct it.

Reply:

Thank you for reviewer’s suggestion. It was the author's carelessness that did not to notice the image resolution problem. The resolution of the illustration figure in the original text has been improved to 600dpi. Relevant modifications have been added to the manuscript.

5.State the research perspective in the form of future work.

Reply:

Thank you for reviewer’s suggestion. In response to your suggestions, I have added a section for prospects and shortcomings in the discussion.

The prospects and shortcomings are as follows:

In this study, we only applied this model to the identification of external quality of peaches of one variety, and it is not universal in the detection of external quality of peach-es of other varieties. Therefore, in the future study, we will use more fruit varieties to realize the detection research of the model, so as to improve the universality of the model. (Lines:518-521)

Reviewer 2 Report

Thank you very much for giving me the opportunity to review this manuscript.

The authors evaluated Research on Defect detection of Kubo peach Based on Hyperspectral Imaging Technology combined CARS-MIV-GA-SVM  method.

General remark

Please follow the writing instructions in the journal template, with the aim of all the text being harmonized with the same way of writing.

Specific remarks

Please rewrite the abstract and limit to the main findings without too much detail

Line 1-79. Please provide the appropriate reference.

Figure 5. Please provide a new one as good peach line is not visible.

Author Response

Research on Defect detection of Kubo peach Based on Hyperspectral Imaging Technology combined CARS-MIV-GA-SVM method

Reviewer: 2 - Comments and Suggestions for Authors

The authors evaluated Research on Defect detection of Kubo peach Based on Hyperspectral Imaging Technology combined CARS-MIV-GA-SVM method.

Reply:

Much appreciate reviewer’s comments. Your encouragement will further strengthen our confidence in the use of spectral non-destructive testing of food products. We sincerely appreciate the reviewer’s efforts in reviewing our manuscript. The manuscript was revised carefully according to the following comments in the marked manuscript. The revised parts are highlighted in yellow in the revised manuscript. Detailed responses are given below.

1 Please follow the writing instructions in the journal template, with the aim of all the text being harmonized with the same way of writing.

Reply:

Thank you for your suggestion. The whole manuscript has been typeset according to the journal template, which will greatly improve the quality of my manuscript.

2-Please rewrite the abstract and limit to the main findings without too much detail.

Reply:

Thanks for the reviewer's suggestion. The main findings of the work in the manuscript are summarized in the abstract section. The revised abstract is as follows:

Abstract: Due to the dark red surface of ripe fresh peaches, its internal injury defects cannot be detected with the naked eye and conventional images. Rapid and accurate detection of fresh peach defects can improve the efficiency of fresh peach classification. The goal of this paper was to develop a non-destructive approach for simultaneously detecting internal injury defects and external injuries of fresh peaches, First, we collected spectral data from 347 Kubo peach samples used hyperspectral imaging technology (900-1700nm) and carried out pretreatment. The competitive adaptive reweighting algorithm (CARS), combination algorithm of CARS and average influence value algorithm (CARS-MIV), combination algorithm of CARS and successive projections algorithm (CARS-SPA), and combination algorithm of CARS and the uninformative variable elimination (CARS-UVE) four methods were used to extract the Characteristic wavelength. Based on the characteristic wavelength extracted by the above method, genetic algorithm optimization support vector machine (GA-SVM) model and least square support vector machine (LS-SVM) model were used to establish classification models. The results show that the combination of CARS and other feature wavelength extraction methods can effectively improve the prediction accuracy of the model when the number of wavelengths is small. Among them, the discriminant accuracy of CARS-MIV-GA-SVM model reaches 93.15%. In summary, hyperspectral imaging technology can accomplish the accurate detection of Kubo peaches defects, and provides feasible ideas for the automatic classification of Kubo peach.

3-Line 1-79. Please provide the appropriate reference.

Reply:

Thank you for reviewer’s reminding. It's the author's mistake. The introduction has been rewritten and the corresponding references supplemented.

4-Figure 5. Please provide a new one as good peach line is not visible.

Reply:

Thank you for reviewer’s suggestion. We have replaced the original image with a higher resolution image. The picture after replacement is as follows:

Figure 5. Average spectral curves of the four types of samples

Reviewer 3 Report

The authors present a study on Kubo peach in order to detect defects using hyperspectral imaging(900-1700nm). The study is interesting and complex, but there are certain aspects regarding typing, and writing that need to be paid more attention to:

-          What is PLS model presented in the Abstract?

-          Rewrite the phrase “We collected spectral data from 347 Kubo peach samples, including 198 good peaches, 149 defective peaches (internal injury, scabs, rot). then Base Line, smoothing Median Filter (SMF), standard normalized variate (SNV), Derivative Gap-Segment (D-GS), Derivative Savitzky-GoLay (D-SG) pretreatment was carried out for the spectral data and compare the accuracy of establishing PLS mode”.

-          The phrase from the Abstract  “The competitive adaptive reweighting algorithm (CARS), the combination of the competitive adaptive reweighting algorithm and the average influence value algorithm (CARS-MIV), the combination of the competitive adaptive reweighting algorithm and the successive projections algorithm (CARS-SPA), the combination of the competitive adaptive reweighting algorithm and the elimination of no information variables (CARS-UVE) four methods were used to extract the Characteristic wavelength, genetic algorithm to optimize the kernel function parameter (gamma) and penalty factor (Cost) of support vector machine (GA-SVM) and least-square support vector machine (LS-SVM) model were used to establish classification models.” must be rewritten, reformulated.

-          The Abstract must be rewritten and reformulated.

-          The introduction is too long. I appreciate the author's efforts to create a complex Introduction, but it is preferable for easy reading and understanding to rearrange and reduce the main ideas of this paragraph.

-          Also, the “Practical application” is interesting, but this paragraph can be introduced in the Introduction section.

-          Line 285, Figure b means Figure 6b?

-          Attention to typing, writing, placement, and compliance with presentation rules everywhere.

-          Lines 186-187, it is appropriate to switch the sentence, “The CARS-MIV algorithm process is shown in Figure 3” with “The MIV algorithm process is as follows:” after this, the enumeration begins.

Author Response

Research on Defect detection of Kubo peach Based on Hyperspectral Imaging Technology combined CARS-MIV-GA-SVM method

Reviewer: 3 - Comments and Suggestions for Authors

The authors present a study on Kubo peach in order to detect defects using hyperspectral imaging(900-1700nm). The study is interesting and complex, but there are certain aspects regarding typing, and writing that need to be paid more attention to:

Reply:

Much appreciate reviewer’s comments. We sincerely appreciate the reviewer’s efforts in reviewing our manuscript. The manuscript was revised carefully according to the following comments in the marked manuscript. The revised parts are highlighted in yellow in the revised manuscript. Detailed responses are given below.

1-What is PLS model presented in the Abstract?

Reply:

Thank you for reviewer’s reminding. It was the author's carelessness that lack of explanation. It is the Partial Least Squares (PLS)(Lines:233-234).

2-Rewrite the phrase “We collected spectral data from 347 Kubo peach samples, including 198 good peaches, 149 defective peaches (internal injury, scabs, rot). then Base Line, smoothing Median Filter (SMF), standard normalized variate (SNV), Derivative Gap-Segment (D-GS), Derivative Savitzky-GoLay (D-SG) pretreatment was carried out for the spectral data and compare the accuracy of establishing PLS mode”.

Reply:

Thank you for reviewer’s reminder. We have rewritten the passage. The revised sentence is:

First, we collected spectral data from 347 Kubo peach samples used hyperspectral imaging technology (900-1700nm) and carried out pretreatment. (Lines:44-45)

3-The phrase from the Abstract  “The competitive adaptive reweighting algorithm (CARS), the combination of the competitive adaptive reweighting algorithm and the average influence value algorithm (CARS-MIV), the combination of the competitive adaptive reweighting algorithm and the successive projections algorithm (CARS-SPA), the combination of the competitive adaptive reweighting algorithm and the elimination of no information variables (CARS-UVE) four methods were used to extract the Characteristic wavelength, genetic algorithm to optimize the kernel function parameter (gamma) and penalty factor (Cost) of support vector machine (GA-SVM) and least-square support vector machine (LS-SVM) model were used to establish classification models.” must be rewritten, reformulated.

Reply:

Thank you for your suggestion. this paragraph has been rewritten, and the revised content is as follows:

The competitive adaptive reweighting algorithm (CARS), combination algorithm of CARS and average influence value algorithm (CARS-MIV), combination algorithm of CARS and successive projections algorithm (CARS-SPA), and combination algorithm of CARS and the uninformative variable elimination (CARS-UVE) four methods were used to extract the Characteristic wavelength.(Lines:45-49)

.4-The Abstract must be rewritten and reformulated.

Reply:

Thank you for reviewer’s suggestion. In the original manuscript, the abstract was indeed unclear. In the revised manuscript, we have rewritten the abstract. The revised abstract is as follows:

Abstract: Due to the dark red surface of ripe fresh peaches, its internal injury defects cannot be detected with the naked eye and conventional images. Rapid and accurate detection of fresh peach defects can improve the efficiency of fresh peach classification. The goal of this paper was to develop a non-destructive approach for simultaneously detecting internal injury defects and external injuries of fresh peaches, First, we collected spectral data from 347 Kubo peach samples used hyperspectral imaging technology (900-1700nm) and carried out pretreatment. The competitive adaptive reweighting algorithm (CARS), combination algorithm of CARS and average influence value algorithm (CARS-MIV), combination algorithm of CARS and successive projections algorithm (CARS-SPA), and combination algorithm of CARS and the uninformative variable elimination (CARS-UVE) four methods were used to extract the Characteristic wavelength. Based on the characteristic wavelength extracted by the above method, genetic algorithm optimization support vector machine (GA-SVM) model and least square support vector machine (LS-SVM) model were used to establish classification models. The results show that the combination of CARS and other feature wavelength extraction methods can effectively improve the prediction accuracy of the model when the number of wavelengths is small. Among them, the discriminant accuracy of CARS-MIV-GA-SVM model reaches 93.15%. In summary, hyperspectral imaging technology can accomplish the accurate detection of Kubo peaches defects, and provides feasible ideas for the automatic classification of Kubo peach.

5-The introduction is too long. I appreciate the author's efforts to create a complex Introduction, but it is preferable for easy reading and understanding to rearrange and reduce the main ideas of this paragraph.

Reply:

Thank you for reviewer’s suggestion. The introduction has been rewritten and reformulated in the revised manuscript and the practical application part is integrated into the introduction, The revised introduction is as follows:

1.Introduction

"Early maturity Okubo Peach" referred to as "Kubo peach", is a kind of premature varieties of peaches, originated in Japan Okubo, fruit type, fruit weight of about 230-280 grams, dense meat, less fiber, more juice, containing soluble solid 16.48%, high-quality, rich nutrition, popular with consumers[1]. In the growing and harvesting process of Kubo peach, due to climate and unavoidable collision in the harvesting process, Kubo peach is prone to defects such as dark wounds, lesions, rot, scars and marks [2-3]. The existence of these defects on the fruit surface will reduce the quality and market value of the fruit. When purchasing fruits, consumers tend to choose fruits with good appearance. However, for external obvious defects such as scab, rot and other peaches can be detected by conventional visual technology, while mature peaches are mostly red, the naked eye cannot identify the damaged peaches, ordinary machine vision cannot identify, which leads to the classification and sorting efficiency of peaches is not high, affecting the commodity value of the fruit, and then affecting the export and sales of Kubo peaches. At present, the peach fruit on the market mainly relies on manual grading, Manual classification has the problems of low efficiency and low accuracy [4]. Therefore, it is of practical value to study a fast and efficient method for the detection of dark wounds and external obvious defects of Kubo peach.

Hyperspectral Imaging is a non-contact image acquisition technology to obtain continuous spectral information of objects in different wavelength ranges. It combines optical and digital image processing technology to provide rich spectral data and spatial information, making the detection and analysis of the surface composition, chemical composition and specific characteristics of the object more accurate and comprehensive. Hyperspectral imaging combined with chemometric approaches is proven to be a powerful tool for the quality evaluation and control of fruits [5,6] as it enables the assessment of internal properties that cannot be inspected with computer vision, including soluble solid content [7], acidity [8], texture [9]. To detect fruit quality defects, such as immaturity [10], bruising [11] etc. Therefore, it is feasible to use hyperspectral imaging technology to detect the dark wound defects and external defects of peaches.

In recent years, hyperspectral imaging technology has been widely applied to the external quality detection of fruits and vegetables, and the researches mainly include: dates, cucumbers, cherries, citrus, apples, peaches [12-22] etc. Most scholars have combined the hyperspectral imaging technology with the related stoichiometry, and have obtained relatively scientific research results. In terms of internal quality detection, Li et al. [13] combined hyperspectral imaging technology with multiple linear regression model to predict the soluble solid content of Hami jujube, and the correlation coefficient of the prediction set reached 0.857. Li. et al [14] combined hyperspectral technology with CARS method to detect cucumber hardness and water loss, two indexes representing cucumber freshness. The final correlation coefficient of PLSR model for hardness was 0.942, and that of PLSR model for moisture was 0.822. Pullanagari et al. [15] Combined hyperspectral imaging technology with partial least squares regression (PLSR) model and Gaussian process regression (GPR) model to detect the hardness and total soluble solids of sweet cherry. Finally, the correlation coefficient of total soluble solids predicted by the GPR model was 0.88. The correlation coefficient of hardness prediction was 0.60. He et al [16] Combined hyperspectral imaging technology with multiplicative scattering correction, Savitzky-Golay and first-order derivative pretreatment methods to establish a PLSR model to predict the water content of dried purple potato, and the correlation coefficient of the final model reached 0.975. In terms of defect detection: Tang et al [17] fitted the hyperspectral imaging data of apples with different damage levels by piecewise nonlinear curves, studied the spectral data of damaged apples within the band range of 386-1016nm, and concluded that the final score detection accuracy of this method reached 97.33%. Xu et al [18] Combined hyperspectral imaging technology with partial least squares regression model to detect the relationship between the damage degree and internal attribute quality of mango, and graded mango according to the damage degree. The final classification accuracy was 77.8%. WANG et al. [19] selected different pretreatment methods to establish an LS-SVM model based on the spectral data of the upper and lower surfaces of citrus yellow dragon leaves. The results showed that the recognition rates of the upper and lower surfaces of citrus leaves were 100% and 92.5%, respectively, when the second derivative was selected as the pretreatment method. Zhang et al [20] Used mean-PC image, improved watershed segmentation algorithm and hyperspectral technology to distinguish normal oranges and defective oranges, and the overall classification accuracy reached 97.73%. Based on mean-PC5 and simple global threshold method, rotten oranges and intact oranges were identified, and the recognition rate reached 100%. Zhang H et al. [21] used hyperspectral imaging technology to detect scab, black spot, root rot and brown disease of citrus, and the final discrimination rate was 94%. Chen Si et al. [22] used hyperspectral imaging technology and band selection method to propose defect region segmentation and recognition algorithms for peach brown rot and scab, and finally the recognition efficiency of brown rot and scab reached 96.9% and 88.4%.

The above studies all use hyperspectral imaging technology to model and analyze the internal quality and external defects of fruits, and achieve good results. The above research mainly used one or more single feature wavelength extraction methods, which has the problem of wavelength redundancy or too little. Wavelength combination method can maximize the useful information of samples and improve the detection accuracy. There are few reports on the use of CARS-MIV combination method to extract characteristic wavelength and establish GA-SVM and LS-SVM classification models to detect peach dark wound defects.

Based on the above problems, this study aims to use hyperspectral imaging technology combined with stoichiometric methods to achieve rapid, efficient, non-destructive and batch identification of Kubo peach. The specific process of this study includes: (1) Collecting Kubo peach fruit with different defect types, and obtaining hyperspectral image data between 900nm and 1700nm. (2) Select the appropriate pretreatment method by establishing PLS model. (3) The CARS-MIV group method was introduced to extract the feature wavelength and compared with other classical feature wavelength extraction methods. (4) The GA-SVM discriminant model was introduced and compared with the classical LS-SVM model to select the best prediction model. The results showed that it was feasible to use hyperspectral imaging technology combined with chemometrics to detect the dark injury and external defects of Kubo peach. The results of this study provide a way for rapid and efficient non-destructive detection of external quality of Kubo peach by using hyperspectral imaging technology combined with a variety of stoichiometric methods. (Lines:17-124)

6-Line 285, Figure b means Figure 6b?

Reply:

Thank you for reviewer’s reminding. It was the author's carelessness that did not indicate (a)(b)(c) in Figure 6, The revised Figure 6 as follows:

Figure 6. Process of feature wavelength extraction by CARS

7-Lines 186-187, it is appropriate to switch the sentence, “The CARS-MIV algorithm process is shown in Figure 3” with “The MIV algorithm process is as follows:” after this, the enumeration begins.

Reply:

Thank you for reviewer’s suggestions. According to your suggestion, we have switched the sentence and have been revised in the manuscript. (Lines:283)
